# Abscisic acid positively regulates rice spikelet closure

**Youming Huang, Qiusheng Xiao, Huilan Zeng, Xiaochun Zeng**\*

School of Life Sciences and Environmental Resources, Yichun University, Yichun, Jiangxi, China

\* xchzeng2013@163.com

## Abstract

To investigate the mechanisms underlying rice glume closure and develop effective regulatory strategies, we examined both fertile (Xingan Zaozhan, Jiazao 70, and Zhenshan 97B) and sterile (Qiyuan S, Yue 4A, and Zhenshan 97A) varieties. ABA and fluridone (FL) were applied to mature panicles at various concentrations to assess closure rates, duration, and endogenous ABA levels in spikelets. RNA sequencing of lodicules was performed to analyze differential expression of genes involved in ABA synthesis and signal transduction during opening and closing phases. Additionally, methyl jasmonate (MeJA) combined with ABA (MeJA+ABA) was applied to female parents in hybrid breeding to evaluate seed setting rate, spikelet cracking rate, and yield increase. ABA treatment significantly accelerated spikelet closure, reduced closure duration, and increased final closure rates in both fertile and sterile lines, with optimal concentrations of 200–400 mg/L. In contrast, FL treatment delayed closure, prolonged the opening-closing duration, and decreased final closure rates. Endogenous ABA levels in palea and lemma were significantly lower during initial opening and closing stages compared to the maximum opening angle. During the Ory-15 min vs. Ory-40 min transition, genes encoding 9-cis-epoxycarotenoid dioxygenase (NCED) and abscisic aldehyde oxidase (AAO) were significantly up-regulated, along with PYR/PYL, SnRK2, and ABF family genes, while PP2C family genes were down-regulated. The combined MeJA+ABA treatment significantly improved glume closure rates and hybrid seed yield. These results demonstrate that ABA promotes rice spikelet closure while FL inhibits it, highlighting ABA's crucial role in hybrid breeding. The MeJA+ABA combination offers a promising strategy to enhance hybrid seed yield and quality.

## 1. Introduction

In rice cultivation, particularly hybrid rice seed production, spikelet closure is as essential as spikelet opening, directly affecting seed yield and quality. During the heading and flowering stages, rice plants face various adverse conditions, including

**Data availability statement:** All relevant data are within the manuscript and its Supporting Information files.

**Funding:** Xiaochun Zeng（XCZ）the National Natural Science Foundation Project (31360297) National Natural Science Foundation of China https://www.nsfc.gov.cn/.

**Competing interests:** The authors have declared that no competing interests exist.

high temperatures, drought, rain, mechanical damage, diseases, pests, and dry-hot winds, all of which impair spikelet closure. Notably, male sterile rice lines commonly exhibit delayed palea/lemma closure, prolonged opening/closing durations, incomplete closure, or even failure to close, resulting in reduced seed filling rates, increased cracking rates, abnormal ovary development, and grain abnormalities. Consequently, these factors compromise seed germination rates, vitality, yield, quality, and overall application value [1,2]. Although flowering synchronization in hybrid rice seed production has been improved by selecting male sterile lines with compatible flowering times and implementing optimized cultivation and chemical regulation practices, the inability to achieve complete spikelet closure in sterile lines remains a critical bottleneck, significantly constraining hybrid seed yield and quality. Therefore, systematic investigation into the initiation factors, regulatory elements, underlying mechanisms, and physical/chemical regulation techniques of rice spikelet closure is imperative.

Rice lodicules, functionally analogous to petals in other plant species [3], are essential for spikelet blooming by absorbing water and expanding, separating the hook between the palea and lemma [4–6]. This swelling induces rupture of cell membranes and parenchyma cell walls due to extreme cellular enlargement. Subsequently, osmotic equilibrium is disrupted, ceasing water uptake, while hydrolytic enzymes released from vacuoles initiate autolysis. This cellular degradation enables water and solutes to translocate from lodicules to the rachilla via vascular conduits, ultimately resulting in lodicule shrinkage and spikelet closure [7–13].

Understanding the dynamics of spikelet opening and closing is fundamental for regulating rice spikelet closure. Similar to stomatal movement in plants, this process involves turgor-driven expansive movements. It occurs rhythmically at specific times of the day and responds to external stimuli such as mechanical damage, light intensity changes, or temperature variations [7]. Previous research [14] has delineated four distinct stages in the opening and closing of rice spikelets: a gradual increase in the angle between the palea and lemma (opening), the maintenance of the maximum angle (blooming), a gradual decrease in the angle (closing), and the attainment of the minimum angle (closed). This systematic classification underscores the independent yet interconnected nature of the "opening and closing" processes.

Previous studies have extensively investigated the opening of rice spikelets [15–18].These techniques for regulating spikelet opening provide new perspectives for overcoming asynchronous flowering between parental lines in hybrid rice seed production and have broad and potential application value.The issue of asynchronous flowering between parents in hybrid rice seed production has been significantly resolved through measures such as selecting combinations of sterile lines and restorer lines with similar flowering times, breeding sterile lines with favorable flowering characteristics, and adopting cultivation and chemical regulation techniques.

However, the inability of the female parent to close open spikelets has emerged as a new bottleneck and a "choke point" in current production, severely restricting the yield and quality of hybrid seeds.So far, research on the closing of rice spikelets has been relatively limited.Youming Huang et al. [19] first reported that ABA promotes

spikelet closure in rice.To validate this effect, and more importantly, to explore its underlying mechanism and identify additional regulatory techniques for rice spikelet closure, in this study, using three fertile and three sterile rice lines, we tested the effects of ABA and fluridone (FL, an ABA biosynthesis inhibitor) on rice spikelet closure. monitored the dynamic levels of endogenous ABA in lodicules during the closure process, and further analyzed the differential expression of key enzyme genes involved in ABA biosynthesis and catabolism, as well as ABA signal transduction-related genes, at the transcriptional level.

Additionally, Xiaochun Zeng et al. found that Jasmonic acid (JA) and Methy Jasmonate (MeJA) can promote the opening of mature rice spikelets [16], and the effect is more pronounced in male sterile lines [17].Youming Huang et al. also found that Abscisic acid (ABA) has a promoting effect on the closing of already opened rice spikelets [19]. So, we applied a methyl jasmonate (MeJA) and ABA mixed solution MeJA+ABA to the female parent during the hybrid rice seed production to evaluate its impact on the hybrid seed setting rate, spikelet cracking rate, and yield increase rate. This comprehensive approach aimed to elucidate the influence of ABA on rice spikelet closure and its potential applications in agriculture.

## 2. Materials and methods

### 2.1. Plants and reagents

Three fertile rice varieties (Xinganzaozhan, Jiazao 70, and Zhenshan 97B) and three sterile rice varieties (Qiyuan S, Yue 4A, and Zhenshan 97A), which are widely used in agricultural production, were selected for this study. The sterile rice lines exhibited characteristic floret-opening phenotypes, including delayed closure, incomplete sealing, and persistent openness of a considerable proportion of spikelets. These materials were sown at the Experimental Demonstration Base of Yichun University growing seasons of 2022 and 2023 (June to September). Standard management practices were applied, and field experiments were conducted along, with panicle sampling. Yichun University's base is characterized by level 1 soil fertility with a profile structure of A-P-W-C, pond-type water irrigation, and clay loam texture. The soil boasted favorable attributes, including 3.5% organic matter content, 0.22% total nitrogen, 7.6 mg·kg$^{-1}$ available phosphorus, 96 mg·kg$^{-1}$ available potassium, 19.5 cmol (+) kg-1 cation exchange capacity, and a pH,6.8. Chemicals utilized in the study were sourced from Sigma, including ABA (CAS number 14375-45-2), FL (CAS number 59756-60-4), and methyl jasmonate (CAS number 39924-52-2).

### 2.2. Methods

**2.2.1. Processing methods.** Rice spikelet opening and closing dynamics are strongly influenced by environmental factors. Gu et al. [15,20] reported that temperatures of 35–40 °C, atmospheres containing over 5% $CO_2$, and saturated $CO_2$ aqueous solutions significantly stimulate spikelet opening. In contrast, low temperatures and respiratory inhibitors inhibited both spikelet opening and closure. Furthermore, Huang et al. [14] identified a "three-basis-point" temperature effect on the closure of open rice spikelets, highlighting that temperatures between 35 and 40°C are optimal for spikelet closure, while temperatures exceeding 45°C or below 20°C hindered closure. Therefore, we conducted our experiments during the peak flowering stages of rice, selecting sunny weather with daily average temperatures surpassing 28°Cand relative humidity exceeding 65%.

Before the experiment, spikelet opening and closing behaviors were monitored 1 ~ 2 days in advance in fields with uniform rice growth to determine the flowering time. ABA solutions were prepared at 0, 50, 100, 200, 400, and 800 mg·L$^{-1}$ [21,22], and FL solutions at 0, 5, 10, 20, 40, and 80 mg·L$^{-1}$ [23,24]. On the experimental day, selected rice plants were uniformly sprayed with the respective solutions at 500 mL per 5 m$^2$, 2 h before flowering, with three replicates.

**2.2.2. Tested characteristics.** In the experimental paddy field, soil fertility and meteorological conditions were consistently maintained to ensure uniformity across rice varieties. Additionally, cultivation and management practices were standardized for each variety to ensure homogeneous growth conditions. To ensure material uniformity, panicles and spikelets were randomly sampled after the spraying treatment.

(1) Spikelet closure rate (%),Spikelet closure rate was determined by recording the number of closed spikelets at 30-min intervals from closure onset until cessation. The rate was calculated as: (number of closed spikelets/ total number of open spikelets at the start) × 100%. Each sample comprised ten panicles, with three replicates per treatment.

(2) Spikelet opening-to-closing duration (min), Opening-to-closing duration was defined as the time interval from initial glume cracking to complete closure. For each treatment, 50 spikelets that opened on the experimental day were selected as one sample, with three replicates.

(3) Endogenous ABA levels in spikelets (ng/g.FW)

Spikelets from the experimental fields of Xingan Zaozhan and Qiyuan S, treated with either 0 mg/L ABA or 0 mg/L FL, were at three different stages on the day of the experiment: when the lemma and palea had just opened, at the maximum lemma–palea angle, and when the lemma and palea closed. Three replicates were performed for each treatment,and immediately frozen into liquid nitrogen for storage quickly.The frozen spikelets (0.9–1.1 g) were ground into a fine powder with a pre-cooled mortar and pestle for 5 min in 2 mL of 80% ethanol. The powder was transferred into a 10 mL centrifuge tube. The mortar was rinsed with 2 mL of 80% ethanol twice and combined in the same tube. The mixture was extracted at 4°C for 4 h or overnight followed by centrifugation at 4,000 rpm for 15 min. The residue was extracted with 1 mL of 80% ethanol at 4°C for 12 h and centrifuged again. The combined supernatant was purified with C18 column, dried under $N_2$ and stored at −20°C.Eendogenous ABA content in spikelets was determined by the indirect enzyme-linked immunosorbent assay (ELISA) method established by the Crop Chemical Control Lab of China Agricultural University.

(4) Differential expression analysis of ABA synthesis and signal transduction-related genes in lodicules

In the Xingan Zaozhan experimental field, spikelets treated with 0 mg/L ABA or 0 mg/L FL were collected at four time points, as previously described [14] on the day of the experiment: when the palea and lemma began to open (Ory-0 min), 15 minutes after opening (Ory-15 min), 40 minutes after opening (Ory-40 min), and when closed (Ory-closure).Ory-0 min vs. Ory-15 min, Ory-15 min vs. Ory-40 min, and Ory-40 min vs. Ory-closure corresponded to the cracking, opening, and closing stages, respectively. Spikelets were rapidly collected and flash-freezing of spikelets were carried out to ensure data integrity. Lodicules were dissected from the spikelets on ice in a freezing laboratory (17 °C) and immediately stored at −80 °C. Each sample was collected in triplicate. In November 2023, all samples were sent to the Wuhan Huada Gene Technology Service Center for RNA sequencing using the Illumina HiSeqTM 2000 high-throughput sequencing platform. Twelve differentially expressed genes involved in ABA biosynthesis and signal transduction were randomly selected for further validation by quantitative real-time PCR (qPCR) using the TransScript II All-in-One First-Strand cDNA Synthesis SuperMix.

(5) Validation of differentially expressed genes by quantitative fluorescence PCR

Selected 15 differentially expressed genes involved in ABA biosynthesis and signaling transduction randomly,using the extracted RNA samples as templates,Reverse transcription was performed using the TransScript II All-in-One First-Strand cDNA Synthesis SuperMix for qPCR(Wuhan Tianyi Huiyuan Biotechnology Co., Ltd.) reagent.Gene expression levels were validated by quantitative real-time PCR (qRT-PCR) using the SYBR Green dye method.The reference gene and its primer sequences used in this study were based on the research by He Yongming et al. [25] on lemma, and OsActin1 was selected as the reference gene.The fold change of gene expression was calculated using the 2-ΔΔCt method.Each sample was set up with three technical replicates.

(6) Application of MeJA+ABA solution in hybrid rice seed production

In order to achieve synchronized and abundant opening of the spikelets of the female parent,and ensure good closure in the seed production of hybrid rice,combining the MeJ and the ABA,the female parent was treated with a spray of

(MeJA+ABA) solution.An area with uniform conditions for both parents (Qiyuan S × Xing'an Zaozhan) was selected in the seed production fields of Jiangxi Xing'an Seed Industry Co., Ltd. The area was divided into six blocks marked with visible markers, each spanning 30 m2. A mixed solution containing the optimal concentrations of MeJA (300 mg/L) and ABA (400 mg/L) for regulating the opening and closing of rice spikelets was prepared. This (MeJA+ABA) solution was uniformly sprayed on the rice panicles of the mother plants at a rate of 1000 mL/30m2 at 8:00 am on a sunny day during the peak flowering period. Three blocks were sprayed as three replicates and three blocks were sprayed with clean water served as the control. Fertilization commenced around 10:30 am. At hybrid seed harvest, 15 panicles were randomly sampled from each block to measure the seed setting rate of the female parent and the spikelet cracking rate of hybrid seeds, and the yield was measured for each treatment, with 3 blocks as 3 replicates.

## 2.3. Data processing

The closure rate, the duration of opening and closing, and the endogenous ABA levels of spikelets were analyzed using Microsoft Excel software and plotted into a column chart.The data in the Figures and tables were the means and standard deviations of three biological sets of data with each biological set came from three technical measurements. Mean values within table columns were analyzed with Duncan new multiple range test. F-tests and multiple comparisons were performed between the opening and closing of spikelets and their endogenous ABA levels.Values in a column with different letters indicate a significant different difference among the times and the levels.The capital letters(A,B,C...) and the lowercase letters(a,b,c...) on the column represent significant differences between floret stages at $p=0.01$ and $p=0.05$, respectively.

Clean reads were aligned to the reference genome using Bowtie2 [26], and gene expression levels were quantified with RSEM [27]. Differentially expressed genes (DEGs) were log2Ratio |log2Ratio| ≥ 1 thresholds [28]. DEGs were functionally annotated by BLAST searches against the NCBI non-redundant (Nr) database (parameters: -p blastx, -e 1e-5, -m 7) and subjected to Gene Ontology (GO) enrichment analysis using Blast2GO (http://www.cytoscape.org/). Additionally, KEGG pathway annotation was performed using BLAST (parameters: -p blastx, -e 1e-5, -m 8) in the Kyoto Encyclopedia of Genes and Genomes (KEGG) database [29]. Metabolic pathways significantly enriched in DEGs were identified with a Q-value ≤ 0.05 threshold. Hybrid seeds from (Qiyuan S × Xing'an Zaozhan) were tested with regard to the seed setting rate, cracking rate, and yield increase rate, and the results were compared in a tabular format.

# 3. Results

ABA is an pivotal plant hormone that critically regulates plant growth and development, as well as,responses to stress. Its functions encompass promoting stomatal closure, regulating seed germination, participating in stress responses, controlling plant growth and development, modulating the aging process of plants, and interacting with other hormones.What about the effect of ABA on the closure of already open rice spikelets? The research results are as follows.

## 3.1. Effect of ABA on the closure of fertile rice spikelets

From 30 to 90 minutes after the initial closure of the first spikelet, the closure rates of Xingan Zaozhan and Jiazao 70 significantly increased with an rising ABA levels, reaching 100% by 120 minutes (Fig 1A-L, 1B-L). Similarly, from 30 to 60 min after the first closure, Zhenshan 97B showed no significant difference in closure rates with ABA concentrations of 200, 400, and 800 mg/L (Fig 1C-L). However, all three concentrations produced significantly higher closure rates than those of the 100, 50, and 0 mg/L treatments, with the closure rate trend being 100 > 50 > 0 mg/L. By 90 min, the closure rate in each treatment reached 100%.

The average opening and closing times of Xingan Zaozhan spikelets are presented in Fig 1A-R ($0.01 < p < 0.05$ between treatments). No significant differences among treatments of 0, 50, and 100 mg/L, or among those of 200, 400, and 800

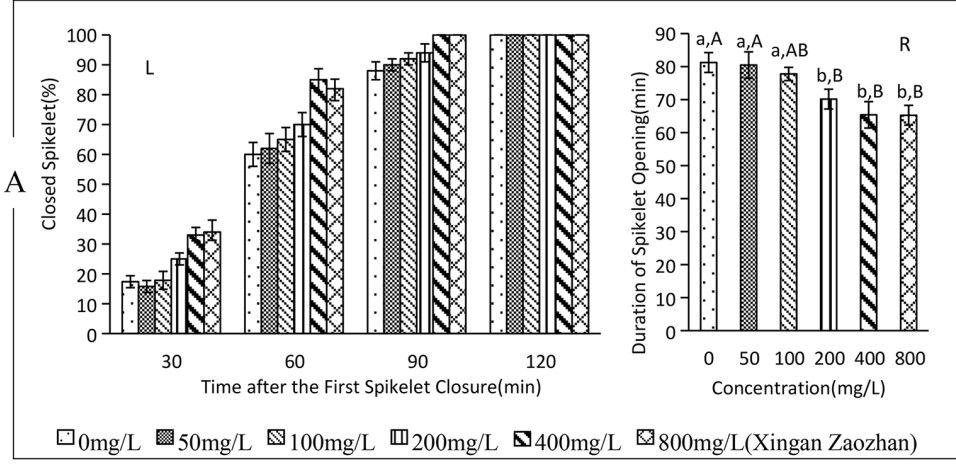

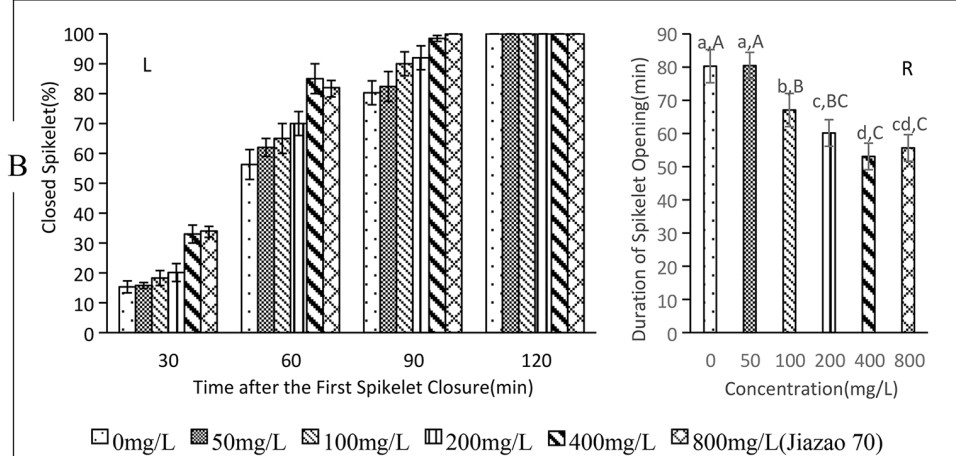

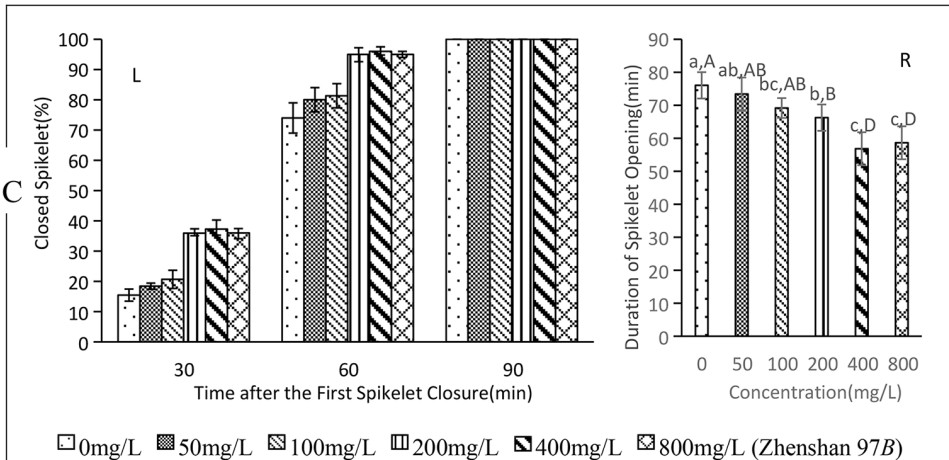

**Fig 1. Effect of ABA on spikelet closure in three fertile rice varieties.** (A) Xingan Zaozhan, (B) Jiazao 70, (C) Zhenshan 97B. L: left figure, R: right figure. Lowercase letters a, b, **c**... indicate significant differences, while Capital letters A, B, **C**... indicate highly significant differences. The concentrations of ABA at 50, 100, 200, 400, and 800 mg/L correspond to 0.19, 0.38, 0.76, 1.51, and 3.03 mM, respectively. The data in this figure are the means and standard deviations of three independent samples.

mg/L. However, closure times for treatments of 200, 400, and 800 mg/L were significantly shorter than those for 0 and 50 mg/L. Compared to the 0 mg/L treatment, the 200, 400, and 800 mg/L treatments shortened closure times by 11.08 min(13.64%), 15.85 min (19.51%), and 15.98 min(19.67%), respectively.

The average opening and closing times of Jiazao 70 spikelets (Fig 1B-R) $p < 0.01$ for all pairwise comparisons. No significant differences among 0 and 50 mg/L treatments, but both exhibited significantly longer times than the others. The difference among 100 and 200 mg/L was significant but not extreme. Treatment with 200 mg/L resulted in significantly longer times compared to with 400 and 800 mg/L, but no significant difference was found among 400 and 800 mg/L. Compared to the time for the 0 mg/L treatment, those for treatments of 100, 200, 400, and 800 mg/L were shortened by 13.22 min (16.47%), 20.14 min (25.08%), 27.21 min (33.89%), and 24.64 min (30.69%), respectively.

The average opening and closing times of Zhenshan 97B spikelets (Fig 1C-R) are presented (0.01 < $p$ < 0.05) across treatments. No significant differences in time between 0 and 50, 100 and 200, and 400 and 800 mg/L. However, times for both 0 and 50 mg/L were significantly longer than those for 200 mg/L, and the time for 200 mg/L was also significantly longer than that for 400 and 800 mg/L treatments. Compared to 0 mg/L, treatments of 100, 200, 400, and 800 mg/L shortened times by 6.88 min (9.05%), 9.80 min (12.89%), 19.22 min(25.27%), and 17.41 min (22.89%), respectively.

These results demonstrate that ABA treatment significantly accelerates the spikelet closure process in fertile rice varieties Xingan Zaozhan, Jiazao 70, and Zhenshan 97B, thereby reducing the duration of opening and closing. The optimal concentration range is 200–400 mg/L.

### 3.2. Effect of ABA on spikelet closure in sterile rice lines

From 30 to 210 min post-initial spikelet closure, ABA treatment induced a significant concentration-dependent increase in closure rates for Qiyuan S, Yue 4A, and Zhenshan 97A (Fig 2A-L, 2B-L, 2C-L). Beyond 210 min, closure rates stabilized with minor fluctuations across all treatments. However, no treatment achieved complete closure (100%) by the experimental endpoint.

The average opening and closing time of Qiyuan S spikelets (Fig 2A-R) are presented, with $p < 0.01$ across treatments. No significant differences were observed among 0, 50, and 100 mg/L, or between 400 and 800 mg/L treatments. However, the times for treatments of 0, 50, and 100 mg/L were significantly longer than those for the 200 mg/L treatment, and the time for 200 mg/L treatment was also significantly longer than that for the 400 and 800 mg/L treatments. Relative to the 0 mg/L treatment, the 200, 400, and 800 mg/L shortened time by 20.7 min (12.84%), 35.45 min(21.98%), and 42.65 min (26.45%), respectively.

The average opening and closing times of Yue 4A spikelets are presented in Fig 2B-R ($p < 0.01$ across treatments). There was a significant but moderate difference among the 0, 50, and 100 mg/L treatments, with no significant difference observed between 400 and 800 mg/L. However, the times for the 0, 50, 100, and 200 mg/L treatments exhibited all significantly longer than those for the 400 and 800 mg/L treatments. Compared to the 0 mg/L treatment, times for 100, 20, 400, and 800 mg/L treatments were shortened by 13.56 min(6.24%), 17.24 min(7.93%), 32.04 min (14.75%), and 37.27 min (17.15%), respectively.

The average opening and closing times of Zhenshan 97A spikelets are presented in Fig 2C-R ($p < 0.01$ across treatments). There was no significant difference between the 0 mg/L and 50 mg/L ABA treatments, whereas there was a significant but moderate difference between the 100, 200, 400, and 800 mg/L treatments. The 50 mg/L treatment exhibited significantly longer times compared to those of the 100, 200, 400, and 800 mg/L treatments. Relative to the 0 mg/L treatment, times for 200 mg/L, 400 mg/L, and 800 mg/L treatments were shortened by 16.56 min(11.25%), 22.16 min (15.05%), and 26.29 min(17.86%), respectively.

These findings demonstrate that ABA treatment significantly accelerates spikelet closure in the male-sterile rice lines Qiyuan S, Yue 4A, and Zhenshan 97A, thereby reducing closure duration and improving the final closure rate. The optimal concentration range was identified as 200–400 mg/L.

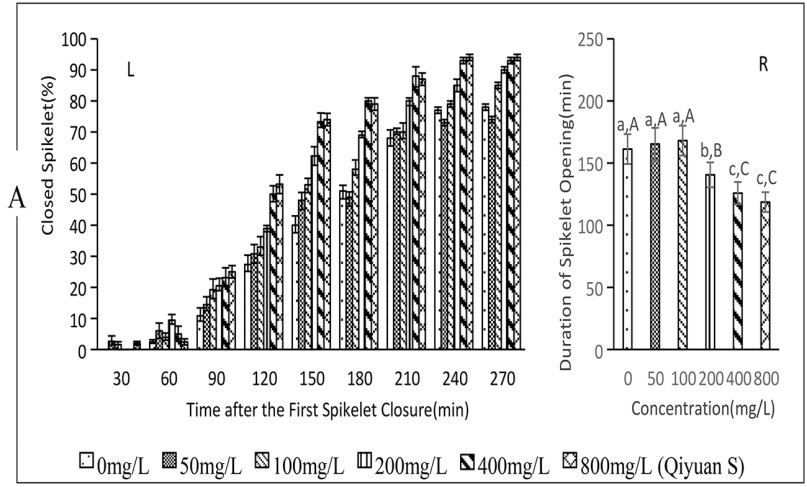

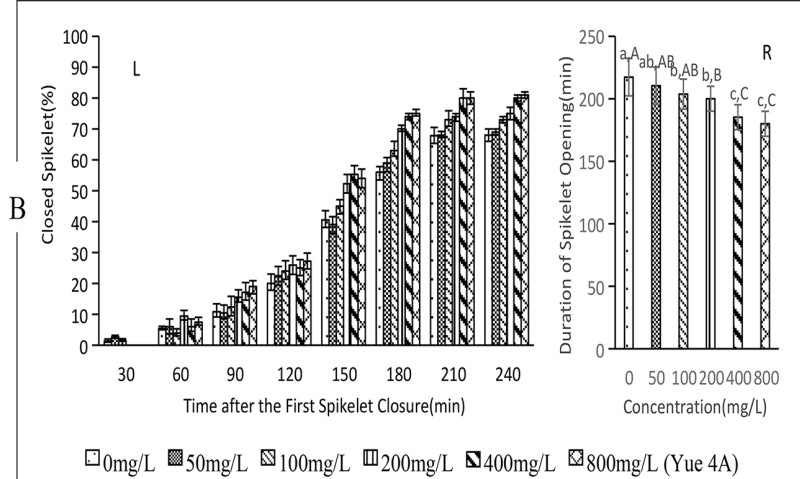

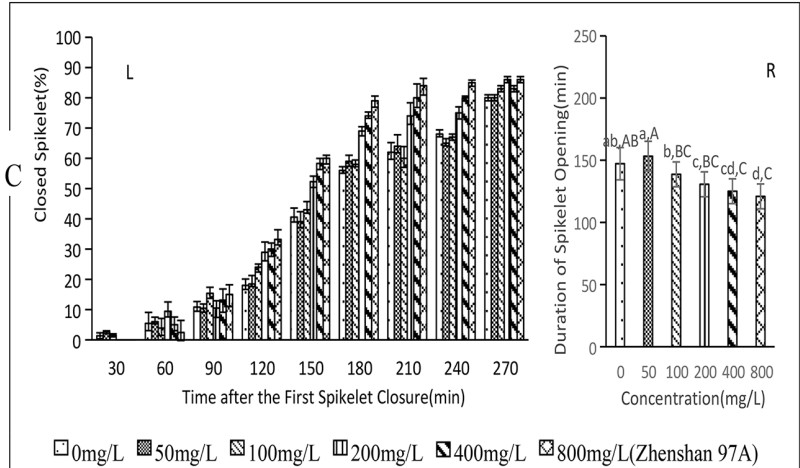

**Fig 2. Effect of ABA on spikelet closure in three sterile rice varieties.** (A) Qiyuan S, (B) Yue 4A, (2C) Zhenshan 97A. L: left figure, R: right figure. Lowercase letters a, b, c... indicate significant differences, while Capital letters A, B, **C**... indicate highly significant differences.The concentrations of ABA at 50, 100, 200, 400 and 800 mg/L correspond to 0.19, 0.38, 0.76, 1.51 and 3.03 mM, respectively. The data in this figure are the means and standard deviations of three independent samples.

### 3.3. Effect of FL on fertile rice spikelet closure

Between 30 and 120 min after the initial closure of the first spikelet, FL treatment resulted in a gradual decline in spikelet closure rates for Xingan Zaozhan, Jiazao 70, and Zhenshan 97B as FL concentration increased (Fig 3A-L, 3B-L, 3C-L). By 150 min, all FL treatments had achieved a 100% closure rate for Xingan Zaozhan spikelets. FL maintained a stable closure rate for Jiazao 70 and Zhenshan 97B spikelets until 150 min or even 180 min, with only control (CK) and low-concentration treatments reaching 100%, indicating that high FL concentration treatment can inhibit spikelet closure.

The average opening and closing times of Xingan Zaozhan spikelets were consistent across treatments (Fig 3A-R; $p<0.01$), accompanied by significant but not extreme differences between 0, 5, 10, and 20 mg/L. No significant difference was observed between 40 and 80 mg/L; however, both 40 and 80 mg/L treatments showed significantly longer times than the 0, 5, and 10 mg/L treatments. Relative to the 0 mg/L treatment, the 10, 20, 40, and 80 mg/L treatments prolonged opening and closing times by 6.95 min (9.10%), 14.02 min(18.37%), 18.97 min (24.85%), and 19.22 min(25.18%), respectively.

The average opening and closing times of Jiazao 70 spikelets (Fig 3B-R) differed significantly across treatments (Fig 3B-R; $0.01<p<0.05$), with significant but not extreme differences among the 0, 5, 10, and 20 mg/L FL treatments. No significant difference was observed between 40 and 80 mg/L; however, both the 40 and 80 mg/L treatments exhibited significantly longer times than the 0, 5, and 10 mg/L treatments. Relative to the 0 mg/L treatment, the 20, 40, and 80 mg/L treatments prolonged the opening and closing times by 12.01 min(15.73%), 18.99 min (24.87%), and 18.58 min (24.33%), respectively.

The average opening and closing times of Zhenshan 97B spikelets differed significantly among treatments (Fig 3C-R; $p<0.01$), with no significant time difference among 0, 5, and 10 mg/L, or between 40 and 80 mg/L FL treatments. However, the 40 and 80 mg/L treatments exhibited significantly longer times than 0, 5, and 10 mg/L treatments. Relative to the 0 mg/L treatment, the 20, 40, and 80 mg/L treatments prolonged the opening and closing times by 17.04 min (23.90%), 21.86 min (30.66%), and 22.75 min (31.91%), respectively.

These findings demonstrate that FL treatment significantly delays spikelet closure in fertile rice varieties Xingan Zaozhan, Jiazao 70, and Zhenshan 97B, leading to a significant prolongation of closure duration and a reduction in final closure rates. The optimal concentration is 20–40 mg/L.

### 3.4. Effect of FL on the closure of spikelets in sterile rice lines

Between 30 and 240 minutes after the initial closure of the first spikelet, the spikelet closing rates (Fig 4A-L, 4B-L, 4C-L) of Qiyuan S, Yue 4A, and Zhenshan 97A under FL treatment decreased noticeably as FL concentration increased. Beyond 240 minutes, the closure rate of spikelets in each treatment stabilized with minimal variation. Even at the conclusion of the experiment, a large number of unclosed spikelets were observed in each treatment.

The average opening and closing times of Qiyuan S spikelets were consistent across treatments (Fig 4A-R; $p<0.01$), with no significant differences among 0, 5, 10, 20,40 and 80 mg/L treatments. However, the 80 mg/L treatment exhibited significantly longer times than the 20 mg/L treatment, which in turn exceeded the 5 mg/L treatment. Relative to the 0 mg/L treatment, the 20, 40, and 80 mg/L treatments prolonged the opening and closing times by 21.44 min(13.48%), 44.76 min (28.15%), and 51.02 min (32.09%), respectively.

The average opening and closing times of Yue4A spikelets (Fig 4B-R) also were consistency, with $p<0.01$ between treatments. There were no significant differences among 0, 5, and 10 mg/L, or between 40 and 80 mg/L treatments. However, the 40 mg/L treatment exhibited significantly longer times than the 20 mg/L treatment, which in turn exceeded the 10 mg/L FL treatment. Compared with the 0 mg/L treatment, the 20, 40, and 80 mg/L treatments prolonged opening and closing times by 71.28 min (34.07%), 104.56 min(49.98%), and 101.59 min(48.56%), respectively.

The average opening and closing times of Zhenshan 97A spikelets (Fig 4C-R; $0.01<p<0.05$) showed no significant differences between the 0 and 5, 10 and 20, 40 and 80 mg/L treatments. Additionally, there was no significant difference

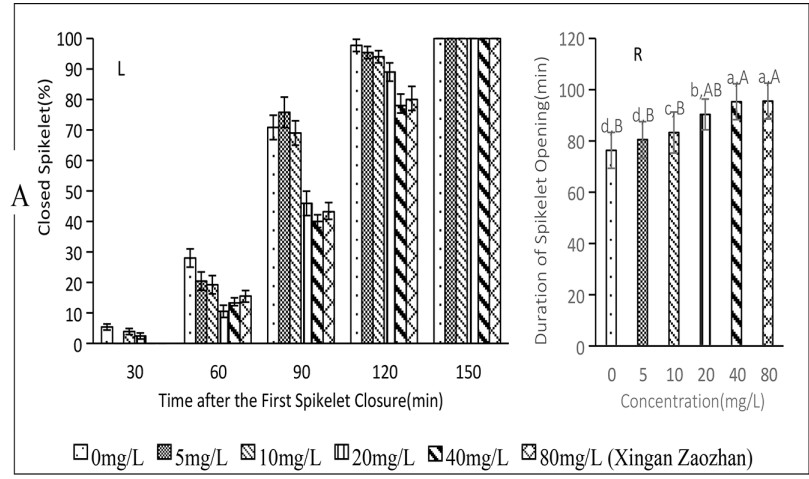

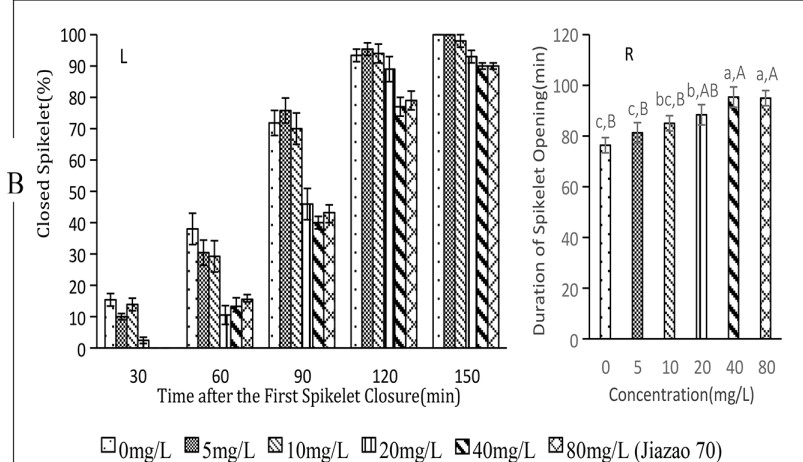

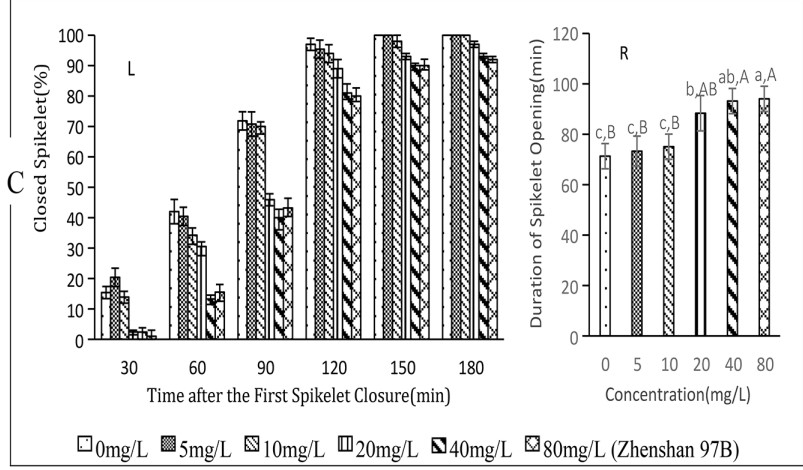

**Fig 3. Effect of FL on spikelet closure in three fertile rice varieties.** (A) Xingan Zaozhan, (B) Jiazao 70, (C) Zhenshan 97B. L: left figure, R: right figure.Lowercase letters a, b, **c**... indicate significant differences, while Capital letters A, B, **C**... indicate highly significant differences.The concentrations of FL at 5, 10, 20, 40, and 80 mg/L correspond to 0.015, 0.030, 0.061, 0.121, and 0.243 mM, respectively. The data in this figure are the means and standard deviations of three independent samples.

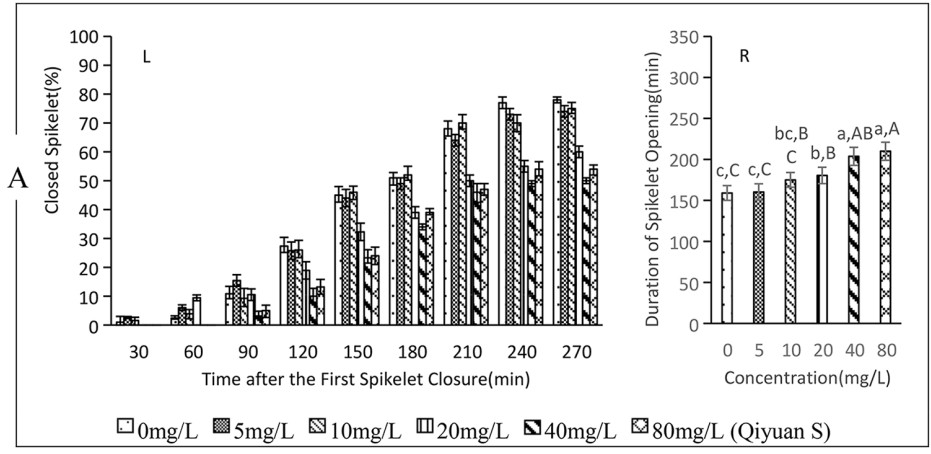

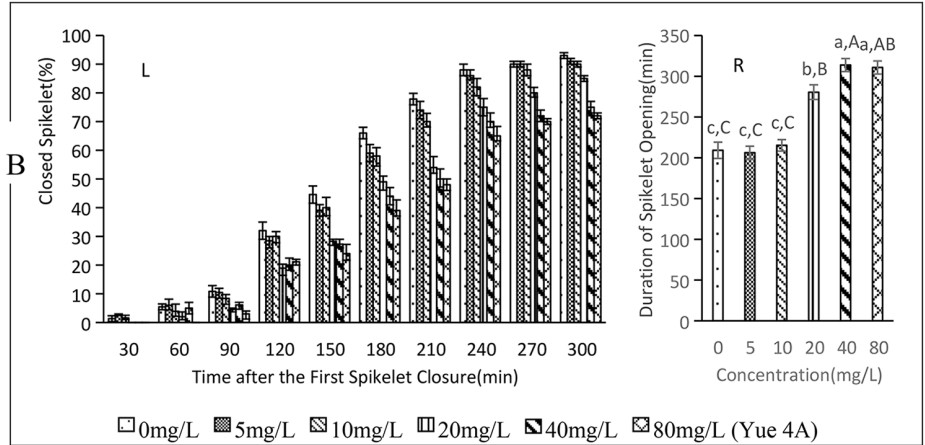

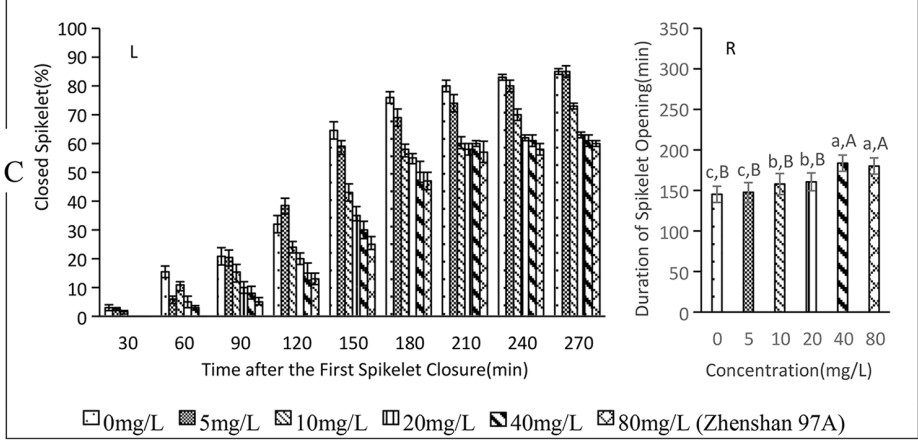

**Fig 4. Effect of FL on spikelet closure in three sterile rice varieties.** (A) Qiyuan S, (B) Yue 4A, (C) Zhenshan 97A. L: left figure, R: right figure. Lowercase letters a, b, **c**... indicate significant differences, while Capital letters A, B, **C**... indicate highly significant differences.The concentrations of FL at 5, 10, 20, 40, and 80 mg/L correspond to 0.015, 0.030, 0.061, 0.121 and 0.243 mM, respectively. The data in this figure are the means and standard deviations of three independent samples.

among 0, 5, 10, and 20 mg/L. However, the 40 and 80 mg/L treatments caused significantly longer opening and closing times than the 0, 5, 10, and 20 mg/L treatments. Relative to the 0 mg/L treatment, the 10, 20, 40, and 80 mg/L treatments prolonged the opening and closing duration by 12.67 min(8.72%), 15.34 min (10.56%), 38.42 min (26.45%), and 34.85 min (23.99%), respectively.

These findings demonstrate that FL treatment significantly delay spikelet closure in sterile rice lines Qiyuan S, Yue 4A, and Zhenshan 97A, leading to a significantly prolonged duration of spikelet opening and closing and reduction in the final closure rate of rice spikelets and inhibitory effects on spikelet closure. The optimal concentration range is 20–40 mg/L.

### 3.5. Dynamics of endogenous ABA levels in spikelets

As shown in Fig 5-L and R, endogenous ABA levels in the spikelets of Xingan Zaozhan and Qiyuan S gradually increased from the onset of opening until the palea and lemma reached their maximum angle, followed by a gradual decrease during closing.For Xingan Zaozhan, significant differences were observed among different opening and closing states ($p < 0.01$), except between the opening and closing moments, which did not differ significantly. However, both states were significantly lower than at the maximum angle stage.In Qiyuan S, the relationship among different states showed a significance level of $0.01 < p < 0.05$. No significant difference was detected between opening and closing times, but the ABA level at maximum angle was significantly higher than during opening and extremely significantly higher than during closing.Compared with the closure stage, ABA levels in the lodicule increased by 26.72% in Xingan Zaozhan and 18.28% in Qiyuan S at maximum opening angle. These results suggest that ABA may act as a critical factor triggering the transition of rice spikelets from opening to closing.

### 3.6. Differentially expressed genes associated with ABA biosynthesis and signaling during spikelet opening and closing

Analysis of ABA biosynthesis pathways and differentially expressed genes (Fig 6, Table 1) revealed that, during the Ory-0 min-vs.-Ory-15 min stage (opening phase), xanthinaldehyde dehydrogenase (Os07g0592100), which converts xanthoxin to abscisic aldehyde, was up-regulated. Concurrently, (+)-abscisic acid 8'-hydroxylase (Os08g0472800), catalyzing ABA degradation to 8'-hydroxyabscisic acid, was also up-regulated.

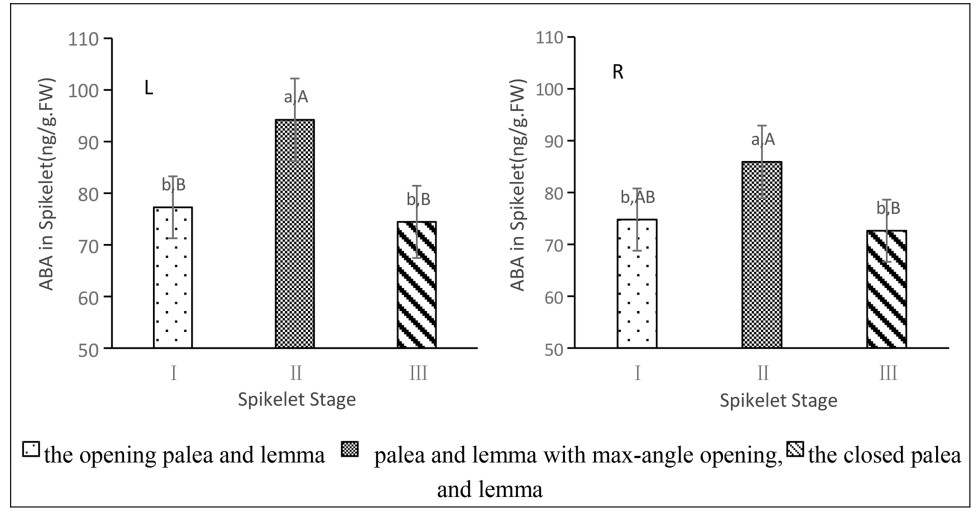

**Fig 5. Endogenous levels of ABA in rice lodicules.** L: left figure, R: right figure. (L: Xinganzaozhan; R: Qiyuan **S**). Lowercase letters a, b, **c**... indicate significant differences, while Capital letters A, B, **C**... indicate highly significant differences.

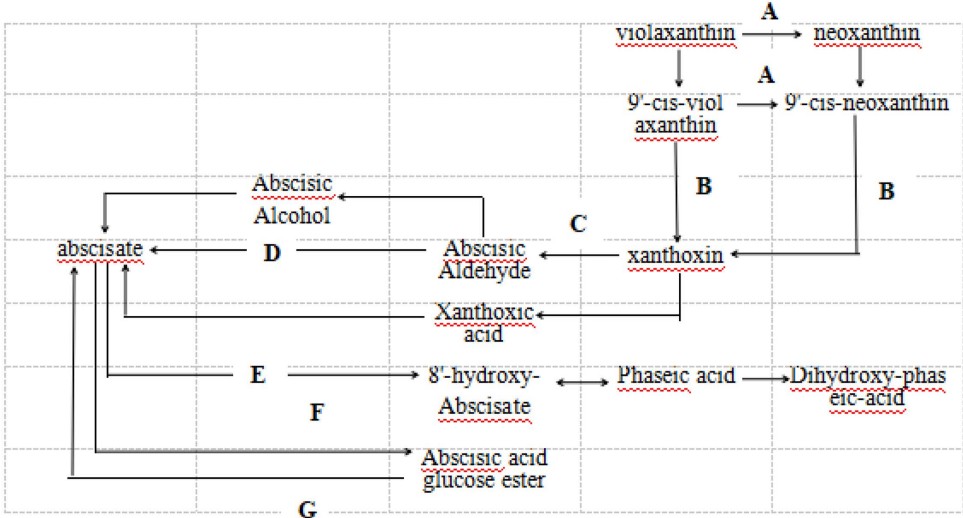

**Fig 6. ABA biosynthesis pathway. (A)** Neoxanthin synthase; **(B)** 9-cis-epoxycarotenoid dioxygenase (NCED); **(C)** Xanthoxin dehydrogenase; **(D)** Abscisic aldehyde oxidase; **(E)** (+)-Abscisic acid 8'-hydroxylase; **(F)** Abscisate β-glucosyltransferase; (G) β-D-glucopyranosyl abscisate β-glucosidase.

**Table 1. Significant DEGs during ABA biosynthesis.**

| *Ory*-0 min-VS-*Ory*-15 min | Ory-15 min-VS-Ory-40 min | Ory-40 min-VS-Ory-closure |
|---|---|---|
| C:Os07g0592100（1.5） | B:Os02g0704000（3.8） | B:Os12g0617400（−2.1） |
| *E:Os08g0472800*(1.4) | B:LOC4342424 （2.1） | B:Os03g0645900（−1.7） |
| | B:Os03g0645900（−1.3） | C:Os04g0179200（−3.0） |
| | C:Os04g0179100（4.9） | C:Os07g0592100（−1.5） |
| | C:Os11g0523110（4.9） | *C:Os07g0664900*（−1.4） |
| | C:Os04g0179200（4.6） | *E:Os09g0457100*（1.1） |
| | C:Os07g0691600（4.6） | *F:Os03g0745100*（−1.8） |
| | C:Os07g0663500（4.4） | *F:Os01g0638000*（−1.0） |
| | C:Os07g0663900（4.2） | |
| | C:LOC4344204（4.0） | |
| | C:Os07g0664000（4.0） | |
| | C:Os07g0663700（4.0） | |
| | C:Os04g0405300（3.3） | |
| | C:LOC4336030（3.1） | |
| | C:Os03g0299200（−1.5） | |
| | C:Os03g0810800（−1.1） | |
| | E:Os09g0457100（1.5） | |
| | E:Os08g0472800（1.3） | |
| | F:Os03g0745100（1.6） | |

During the Ory-15 min-vs.-Ory-40 min stage (opening phase with decreasing palea-lemma angle), multiple genes were up-regulated, including β-carotene hydroxylase (Os10g0533500), which catalyzes zeaxanthin synthesis, and FAD-dependent urate hydroxylase (Os04g0423100), involved in zeaxanthin-to-antheraxanthin and antheraxanthin-to-violaxanthin conversions. Concurrently, zeaxanthin epoxidase (ZEP, Os02g0704000) and 9-cis-epoxycarotenoid dioxygenase (NCED, LOC4342424), which cleave 9-cis-violaxanthin and 9'-cis-neoxanthin to xanthinaldehyde, were up-regulated. Furthermore, several xanthinaldehyde dehydrogenase genes (Os04g0179100, Os11g0523110, Os04g0179200, Os07g0691600, Os07g0663500, Os07g0663900, LOC4344204, Os07g0664000, Os07g0663700, Os04g0405300, and LOC4336030), converting xanthinaldehyde to abscisic aldehyde, were up-regulated.

During the Ory-40 min-vs.-Ory-closure stage (closure phase), β-carotene hydroxylase (Os03g0125100), which catalyzes zeaxanthin synthesis, was down-regulated. Similarly, FAD-dependent urate hydroxylase genes (Os03g0153500 and Os07g0491900), involved in zeaxanthin-to-antheraxanthin and antheraxanthin-to-violaxanthin conversions, were down-regulated. In addition, 9-cis-epoxycarotenoid dioxygenase genes (Os12g0617400 and Os03g0645900), catalyzing the cleavage of 9-cis-violaxanthin and 9'-cis-neoxanthin to xanthinaldehyde, were down-regulated. Furthermore, xanthinaldehyde dehydrogenase genes (Os04g0179200, Os07g0592100, and Os07g0664900), converting xanthinaldehyde to abscisic aldehyde, were down-regulated. Conversely, ABA 8'-hydroxylase (Os09g0457100), which promotes ABA degradation, was up-regulated.

Analysis of ABA signal transduction pathways and differentially expressed genes (Fig 7, Table 2) revealed no significant changes in gene expression across families during the Ory-0 min-vs.-Ory-15 min phase. However, during the Ory-15 min-vs.-Ory-40 min phase, differential expression was detected. Specifically, PYL4 (Os03g0297600), an ABA receptor from the PYR/PYL family, was up-regulated, whereas three PP2C family phosphatases—PP2C66, PP2C8, and PP2C30 (Os08g0500300, Os01g0656200, and Os03g0268600)—were down-regulated. Concurrently, serine/threonine protein kinase SAPK6 (Os02g0551100) from the SnRK2 family was up-regulated. This expression pattern is consistent with the dual negative regulatory mechanism of ABA signaling: PYL4-mediated inhibition of PP2Cs relieves suppression of SnRK2 kinases, leading to activation of downstream targets. Indeed, up-regulation of light-induced protein CPRF2 (Os12g0601800) and bZIP transcription factor TRAB1 (Os09g0456200) in the ABF family was observed, likely resulting from SnRK2-mediated phosphorylation of transcription factors and membrane proteins.During the Ory-40 min-vs.-Ory-closure phase, additional transcriptional changes occurred. Notably, 40S ribosomal protein S6 (Os07g0622100), bZIP transcription factor TRAB1 (Os09g0456200), G-box binding factor 4 (Os01g0867300), and ABA insensitive 5-like protein 2 (Os06g0719500) were all up-regulated. These findings suggest that ABA signal transduction is transcriptionally enhanced during the Ory-15 min-vs.-Ory-40 min phase.

### 3.7. Validati of differentially expressed genes by qRT-PCR

To validate the RNA-Seq results, 15 DEGs involved in ABA biosynthesis and signaling (RESM > 20) were randomly selected for qRT-PCR confirmation. Gene-specific primers and the reference gene are listed in Table 3. Quantitative real-time PCR with fluorescence detection was performed as described in the Methods. Although fold changes differed numerically between qRT-PCR and RNA-Seq, expression trends were highly consistent (Fig 8), confirming the reliability of the transcriptome data.

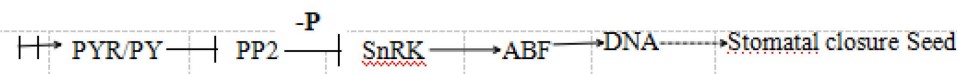

**Fig 7. ABA signal transduction.** PYR/PYL: ABA receptor PYR/PYL family; PP2C: protein phosphatase 2C; SnRK2: serine/threonine-protein kinase SRK2; ABF: ABA-responsive element binding factor.

**Table 2. Significant DEGs involved in ABA signal transduction.**

| Ory-0 min-VS-Ory-15 min | Ory-15 min-VS-Ory-40 min | Ory-40 min-VS-Ory-closure |
|---|---|---|
| | PYR/PYL: | SnRK2: |
| | Os03g0297600（5.3） | Os07g0622100（1.3） |
| | PP2C: | ABF: |
| | Os09g0325700（1.3） | Os09g0456200（1.7） |
| | Os08g0500300（−3.2） | Os01g0867300（1.6） |
| | Os01g0656200（−2.8） | Os06g0719500（1.3） |
| | Os03g0268600（−2.0） | |
| | SnRK2: | |
| | Os02g0551100（3.9） | |
| | Os05g0400600（−1.4） | |
| | ABF: | |
| | Os12g0601800（4.8） | |
| | Os09g0456200（2.3） | |

**Table 3. Primers of qRT-PCR for the 15 selected genes.**

| No. | Gene ID | Forward primer | Reverse primer |
|---|---|---|---|
| 1 | Os07g0592100 | GCTGCTGATGGATGGGAAGAA | CACCACAAAATCCTCCCTCTCA |
| 2 | Os08g0472800 | CAATGCAACCACCTGCACAA | TTGCCGCTTACAAATAGACTCC |
| 3 | Os02g0704000 | GATGCTACTGAAAGAAGGTACACC | CATCCCACCTCACCAACAAAC |
| 4 | LOC4342424 | GCGAGCGCCTCGAGATCCTC | CGGTGGTGACCGAGAGGTAGTT |
| 5 | Os04g0179100 | GTGTGACTTCTACGGGGTGTTC | TCCATCGTCTCCGTCCAAGA |
| 6 | Os11g0523110 | ACTCCAAGTCATCACTGCTGC | TCTGGGTCCTCAAAAGGGTAA |
| 7 | Os04g0179200 | CGGTTTGCTGACTTGATTGAG | GTCAGCCCAACCAGCATAGT |
| 8 | Os07g0691600 | ACTCCTCTTCGCCTGTTCGACT | CTCCACATCCTCCCTATCGCA |
| 9 | Os07g0663500 | CTTCCACGGCACCTTCATCAA | GATTGGAAGGTGTTTTGGAATGA |
| 10 | Os07g0663900 | GGTGATGTCCAGCCTCGGGAT | GTGCCGAACATTTCCGTGACC |
| 11 | LOC4344204 | CATGTCATCCACACGAATTTTCTT | GCATCCAACCGAGACATCAAC |
| 12 | Os07g0664000 | ACATCTGTAGCAGGCAAGGGC | TTACCACAGCATCACAAGGCA |
| 13 | Os12g0617400 | CCTTTACTTTCTATTTCAACACAGC | TCACACACTTCTGCTGTAATAATCA |
| 14 | Os03g0645900 | AACAGTCAGATGATGATGGCGA | GGCTGATTCCCGATTTGACA |
| 15 | Os04g0179200 | CTGAGAGATGACCTGGAGTACGT | CAATGGCTTCAGACCCTTTCA |
| 16 | OsActin1 | TCTCTGTATGCCAGTGGTCGTA | GCCGTTGTGGTGAATGAGTAA |

## 3.8. Application of MeJA and ABA in hybrid rice seed production

To confirm the promotive effect of ABA on the opening and closing of mature spikelets in rice, Jiangxi Xingan Seed Industry Co., Ltd. was tasked with conducting a regulatory experiment on the opening and closing of maternal spikelets in (Qiyuan S×Xingan Zaozhan) seed production using MeJA+ABA. This treatment has been reported to significantly promote spikelet opening and closing [16]. In the hybrid seed testing (Table 4 and Fig 9), the (MeJA+ABA) treatment led to a 26.99% reduction in spikelet cracking rate and a 26.11% increase in yield compared to CK. These results signify a

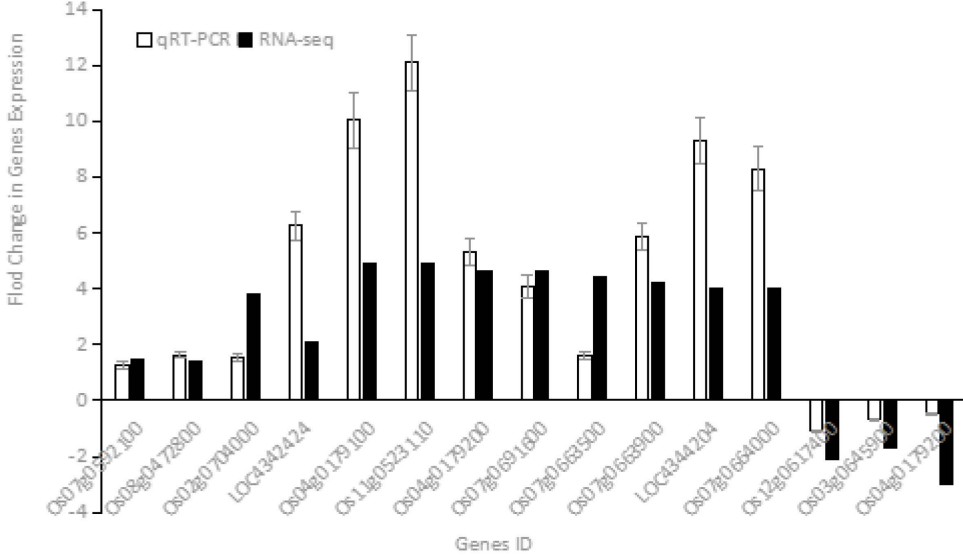

**Fig 8. Comparison of the differentially expressed genes between qRT-PCR and RNA-seq.**

significant enhancement in both the yield and quality of commercial seeds. Thus, (MeJA+ABA) demonstrates substantial potential for enhancing hybrid rice seed production.

## 4. Discussion

This study selected three fertile and three sterile rice varieties (among the sterile varieties, there were both two-line and three-line types, both indica and japonica types) as experimental materials, aiming to understand the universality of ABA's effect in promoting rice floret closing.Figs 1-4 indicated,despite variations among varieties, the overall response pattern was consistent.

Understanding the impact of endogenous ABA on rice spikelet closure is essential for elucidating the regulatory mechanisms underlying ABA-mediated rice spikelet closure. To clarify this effect, we employed FL (Figs 3 and 4), which inhibits the biosynthesis of carotenoid, a precursor substance of ABA, thus exerting physiological effects opposite to ABA [30–32]. For instance, while ABA enhances the cold resistance of winter wheat, FL diminishes it [33,34]. ABA promotes dormancy and inhibits seed germination, whereas FL breaks dormancy and facilitates quick seed germination.Exogenous FL soaking not only enhances the germination rate of rice but also substantially shortens the germination process of rice seeds [35]. Additionally, nordihydroguaiac acid (NDGA) serves as another inhibitor of ABA biosynthesis in rice [36,37], while the plant growth inhibitor 5-isopropyl-2methyl-4i (pyridinecarboxyxy) phenyltrimethylammonium chloride (AMO-1618) promotes ABA synthesis by inhibiting the activity of part A of kaurene oxidase and squalene oxide cyclase, thus inhibiting the synthesis of GA and some steroids [38]. Investigating the effects of ABA biosynthesis promoters and inhibitors on rice

**Table 4. (MeJA+ABA) administration on female parent spikelet closure (QiyuanS/Xinganzaozhan).**

| Treatment | Total Grains/ Panicles | Solid Grains/ Panicles | Setting Seeds (%) | In Solid Grains/Panicles | | | Production (kg/667m²) | Yield Increase (%) |
|---|---|---|---|---|---|---|---|---|
| | | | | Glume Dehiscence Grains | Glume Closure Grains | Closed Grain (%) | | |
| MeJA+ABA | 186.4 | 104.6 | 56.12 | 26.7 | 77.9 | 74.47 | 327.45 | 26.11 |
| CK(Water Equivalent) | 189.1 | 107.2 | 56.69 | 56.3 | 50.9 | 47.48 | 259.66 | |

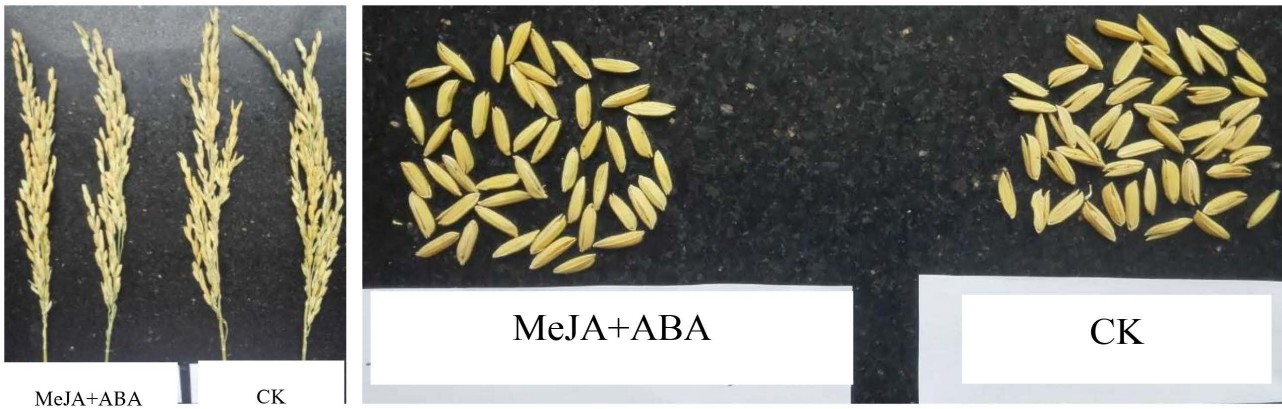

**Fig 9. Influence of (MeJA+ABA) on female parent spikelet closure (Qiyuan S/Xingan Zaozhan).**

spikelet closure not only sheds light on the impact of endogenous ABA but also validates the influence of exogenous ABA. Furthermore, it may offer novel regulatory strategies for spikelet closure. Although this study did not examine the effects of NDGA and AMO-1618 on rice spikelet closure, we aim to address this in future research.

The opening and closing of rice spikelets comprise four distinct stages: cracking, opening, closing, and closed. Our findings reveal that endogenous abscisic acid (ABA) levels increase significantly in spikelets when the opening angle between the palea and lemma is maximal, suggesting that ABA may play a pivotal role in initiating the transition from the opening to the closing stage of spikelets. As plant physiological processes result from the combined action of multiple hormones, it is essential to understand the effects of mutual promotion, antagonism, induction, and feedback. For instance, the external application of ABA can decrease the endogenous $GA_3$ content in young sugarcane leaves and increasing ABA content and the ABA/$GA_3$ ratio, thereby enhancing the cold resistance of sugarcane seedlings [39,40]. Similarly, spraying exogenous ABA or GA significantly increases the corresponding ABA or GA content in wheat grains, leading to a significant rise in IAA and CTK content in wheat grains during the middle and late stages of grain filling [41,42]. However, the present study solely examined endogenous ABA dynamics (Fig 5) and lacked insight into hormones such as IAA, MeJA, and GA. Further investigation is warranted in this regard.

MeJA is currently recognized as the most potent inducer of floret opening, effective in various graminaceous plants including rice, wheat, and sorghum [16]. It promotes floret opening by enhancing ATPase and CAT activities in rice lodicules, facilitating the accumulation of osmotic regulatory substances (such as soluble sugars and ascorbic acid), reducing $H_2O_2$ concentration, enabling lodicule water absorption and swelling, and ultimately driving floret opening [9]. The present study demonstrates that exogenous ABA treatment significantly increases the floret closing rate, shortens floret opening duration, and strongly promotes floret closure. Although MeJA and ABA possess independent biosynthetic pathways representing a metabolic parallel relationship (Fig 6), cross-talk exists between them: ABA→SAPK10 (SnRK2 family kinase) → bZIP72 (transcription factor) → AOC gene expression→increased JA/MeJA synthesis. Upon autophosphorylation, SAPK10 phosphorylates bZIP72, enhancing its protein stability and DNA-binding capacity, thereby promoting AOC promoter activit y [53]. Future research should further elucidate the spatiotemporal specificity mechanisms underlying ABA-JA metabolic cross-regulation during floret opening, particularly the activity status of the SAPK10-bZIP72-AOC module in lodicule cells.

The endogenous ABA level primarily depends on ABA biosynthesis and degradation [43–45]. At the Ory-15 min-vs.-Ory-40 min stage, that is, the transition from the open stage to the closed stage of florets.The genes encoding three key enzymes are all extremely significantly up-regulated: zeaxanthin epoxidase (ZEP), which catalyzes the cyclization

of zeaxanthin to violaxanthin; 9-cis-epoxycarotenoid dioxygenase (NCED), which catalyzes the cleavage of 9-cis-neoxanthin aldehyde to produce the C15 intermediate xanthoxin; and ABA aldehyde oxidase (AAO), which catalyzes the oxidation of ABA aldehyde to ABA.This perfectly matches the measured endogenous ABA levels and the observed effect of ABA in promoting the closing of previously opened rice florets(Fig 6 and Table 1). In future studies, we will strengthen the investigation of the activities of key enzymes (ZEP, NCED, AAO, and CYP) involved in ABA synthesis and degradation in lodicule tissues during rice floret closure, and analyze their impacts on endogenous ABA levels in lodicules.

PYR/PYL/RCAR serves as the most upstream receptors protein in ABA signal transduction and can recognize and initiate ABA signal transduction [46–49]. The PYR/PYL/RCAR protein negatively regulate PP2C phosphatases, which subsequently negatively regulates SnRK2 (SnRK2.2/2.3/2.6), thus forming a double-negative regulatory module in the ABA signaling response. Under normal conditions, maintains SnRK2 in an inactive state through dephosphorylation, keeping ABA signaling quiescent. Once the ABA signal emerges, ABA binds to PYR/PYL/RCAR and promotes its interaction with PP2C, inhibiting PP2C phosphatase activity and alleviates PP2C-mediated repression of SnRK2. Upon activation, SnRK2 phosphorylates downstream transcription factors like ABI5, initiating ABA signaling responses [50–52]. In the Ory-15 min versus Ory-40 min comparison (Fig 7 and Table 2), the PYL4 gene (Os03g0297600) from the PYR/PYL family is up-regulated, while the phosphoprotease genes (Os08g0500300, Os01g0656200, and Os03g0268600) from the PP2C family are down-regulated. SAPK6 (Os02g0551100) from the SnRK2 family and CPRF2 (Os12g0601800) and TRAB1 (Os09g0456200) from the ABF family are up-regulated. This signal transduction expression pattern is essentially consistent with the expression pattern of ABA synthase and endogenous ABA levels.

During the critical transition period from floret opening to closing, the β-carotene hydroxylase, the FAD-dependent urate hydroxylase, ZEP, NCED, and the xanthinaldehyde dehydrogenase genes were all up-regulated, indicating elevated endogenous ABA levels in the floret, which is consistent with our measured results (Fig 5). Concurrently,the ABA receptor PYL4 was up-regulated, whereas three PP2C family phosphatases—PP2C66, PP2C8, and PP2C30 were down-regulated,and the SAPK6 from the SnRK2 family was up-regulated(Fig 7, Table 2). This expression pattern is consistent with the dual negative regulatory mechanism of ABA signaling,and also consistent with the ABA-promoted closure of opening florets observed in this study.Although RNA-seq sequencing identified a large number of DEGs related to ABA biosynthesis and degradation as well as ABA signal transduction, such as xanthoxin dehydrogenase genes and phosphatase genes in the PP2C family, regrettably, this study did not validate their functions (e.g., mutant analysis, transgenic plants). In future research, we will conduct in-depth studies by integrating proteomics.

Preliminary experiments were also conducted to assess the conservation of ABA on spikelet closure in Poaceae plants like maize, wheat, and sorghum. Results demonstrated that exogenous ABA application can significantly accelerated spikelet closure and reduced its duration. Building upon the established role the influence of MeJA on encouraging the opening of rice spikelets [16], a versatile floral timing regulator for Poaceae plants, predominantly consisting of (MeJA+ABA) solution, could be devised(Table 4 and Fig 9). This regulator might be employed to modulate the flowering timing of hybrid seed production and breeding in Poaceae plants, thereby reducing the rate of spikelet cracking rates. This advancement holds substantial potential for enhancing both the yield and quality of hybrid seeds.

To elucidate the molecular mechanisms of ABA on rice spikelet closure and develop effective regulatory strategies, future studies should focus on various aspects. First, there is a need to intensify research on the impacts of ABA structural analogs, synthetic precursors, metabolic promoters, and biosynthetic inhibitors on rice glume closure. Second, it is essential to explore the crosstalk among multiple phytohormones, the activities of key enzymes involved in ABA metabolic pathways, and the dynamics of ion concentrations in pulp cells during spikelet closure. Third, a comprehensive study is warranted on the functional expression of critical genes implicated in ABA signal transduction during spikelet closure.

 

## 5. Conclusions

Our results demonstrate that ABA (abscisic acid) promotes rice spikelet closure, whereas FL inhibits this process. These findings suggest that ABA functions as an initiating factor for spikelet closure. In the seed production of hybrid rice, the combination of (MeJA+ABA) exhibits substantial potential for enhancing the yield and quality of hybrid seeds.MeJA and ABA are the main components of the "Multifunctional Spikelets Regulator for the Female Parent" that we have developed. However, attention should also be paid to the effects of the (MeJA+ABA) solution spray treatment on the photosynthetic capacity of maternal leaves and the stigma viability.

## Supporting information

**S1 Fig. Effect of ABA on spikelet closure in three fertile rice varieties. (A) Xingan Zaozhan, (B) Jiazao 70, (C) Zhenshan 97B.**
(DOC)

**S2 Fig. Effect of ABA on spikelet closure in three sterile rice varieties. (A) Qiyuan S, (B) Yue 4A, (C) Zhenshan 97A.**
(DOC)

**S3 Fig. Effect of FL on spikelet closure in three fertile rice varieties. (A) Xingan Zaozhan, (B) Jiazao 70, (C) Zhenshan 97B.**
(DOC)

**S4 Fig. Effect of FL on spikelet closure in three sterile rice varieties. (A) Qiyuan S, (B) Yue 4A, (C) Zhenshan 97A.**
(DOC)

**S5 Fig. Endogenous levels of ABA in rice lodicules.**
(DOC)

**S6 Fig. ABA biosynthesis pathway.**
(DOC)

**S7 Fig. ABA signal transduction.**
(DOC)

**S8 Fig. Comparation of the differentially expressed genes between qRT-PCR and RNA-seq.**
(DOC)

**S9 Fig. Influence of (MeJA + ABA) on female parent spikelet closure (Qiyuan S/Xingan Zaozhan).**
(DOC)

## Author contributions

**Conceptualization:** Youming Huang, Xiaochun Zeng.

**Data curation:** Youming Huang, Qiusheng Xiao, Huilan Zeng.

**Formal analysis:** Youming Huang, Qiusheng Xiao, Huilan Zeng.

**Funding acquisition:** Xiaochun Zeng.

**Investigation:** Youming Huang, Qiusheng Xiao, Huilan Zeng.

**Methodology:** Youming Huang, Xiaochun Zeng.

**Project administration:** Youming Huang, Xiaochun Zeng.

**Resources:** Youming Huang, Qiusheng Xiao, Huilan Zeng.

**Software:** Qiusheng Xiao, Huilan Zeng.

**Supervision:** Youming Huang.

**Validation:** Qiusheng Xiao, Huilan Zeng.

**Visualization:** Youming Huang.

**Writing – original draft:** Youming Huang.

**Writing – review & editing:** Youming Huang, Xiaochun Zeng.

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
