## [Decision Letter · Decision Letter 0]

23 Jan 2026

PONE-D-25-60666Abscisic acid positively regulates rice spikelet closurePLOS One

Dear Dr. Huang,

Thank you for submitting your manuscript to PLOS ONE. After careful consideration, we feel that it has merit but does not fully meet PLOS ONE’s publication criteria as it currently stands. Therefore, we invite you to submit a revised version of the manuscript that addresses the points raised during the review process. While the reviewers acknowledge the potential significance of your study, they have raised several important concerns that currently limit the strength and impact of your conclusions. These concerns include requests for additional experimental validation, clarification of data interpretation, and improvements in the presentation and discussion of results. Therefore, we ask that you revise your manuscript thoroughly to address all points raised.

We look forward to receiving your revised manuscript.

Kind regards,

Keqiang Wu, Ph.D

Academic Editor

PLOS One

Journal Requirements:

“National Natural Science Foundation Project (31360297)”

“Xiaochun Zeng（XCZ）the National Natural Science Foundation Project (31360297) National Natural Science Foundation of China https://www.nsfc.gov.cn/”

“Xiaochun Zeng（XCZ）the National Natural Science Foundation Project (31360297) National Natural Science Foundation of China https://www.nsfc.gov.cn/”

6. We note that your Data Availability Statement is currently as follows: “All relevant data are within the manuscript and its Supporting Information files.”

7. PLOS requires an ORCID iD for the corresponding author in Editorial Manager on papers submitted after December 6th, 2016. Please ensure that you have an ORCID iD and that it is validated in Editorial Manager. To do this, go to ‘Update my Information’ (in the upper left-hand corner of the main menu), and click on the Fetch/Validate link next to the ORCID field. This will take you to the ORCID site and allow you to create a new iD or authenticate a pre-existing iD in Editorial Manager.

Reviewers' comments:

Reviewer's Responses to Questions

**Comments to the Author**

1. Is the manuscript technically sound, and do the data support the conclusions?

Reviewer #1: Yes

Reviewer #2: Yes

Reviewer #3: Partly

2. Has the statistical analysis been performed appropriately and rigorously?

Reviewer #1: Yes

Reviewer #2: Yes

Reviewer #3: No

3. Have the authors made all data underlying the findings in their manuscript fully available?

Reviewer #1: No

Reviewer #2: Yes

Reviewer #3: Yes

4. Is the manuscript presented in an intelligible fashion and written in standard English?

Reviewer #1: Yes

Reviewer #2: Yes

Reviewer #3: No

5. Review Comments to the Author

Reviewer #1: Overall Recommendation:Major Revision

General Comments:

This study investigates the role of abscisic acid (ABA) in regulating rice spikelet closure using physiological, molecular, and field application approaches. The topic is of significant agronomic importance, especially for hybrid rice seed production. The experimental design is generally sound, and the combination of hormone treatments, RNA-seq, and field trials provides a comprehensive approach. However, several issues need to be addressed before the manuscript can be considered for publication.

Major Comments:

Lack of Statistical Details:

While the manuscript mentions statistical tests (e.g., F-tests, multiple comparisons), it does not provide sufficient details on the statistical methods used, such as the specific type of ANOVA, post-hoc tests, or sample sizes for each experiment. Please include these details in the Methods section and ensure that all figures include clear indications of statistical significance (e.g., asterisks with defined p-values).

RNA-Seq Data Availability:

The manuscript states that "all relevant data are within the manuscript and its Supporting Information files." However, RNA-seq data are not typically included in the main text or supplementary files. The raw or processed RNA-seq data should be deposited in a public repository (e.g., NCBI GEO or SRA), and the accession number must be provided.

Hormonal Interactions:

The study focuses on ABA but also involves MeJA. However, the interaction between ABA and other hormones (e.g., IAA, GA) is only briefly discussed. Given the complexity of hormonal regulation, more discussion or data on how ABA interacts with other hormones during spikelet closure would strengthen the manuscript.

Mechanistic Insight:

While RNA-seq identified DEGs related to ABA synthesis and signaling, functional validation (e.g., mutant analysis, transgenic plants) is lacking. The authors should acknowledge this limitation and discuss how future studies could address it.

Field Application Data:

The field trial results are promising, but the sample size and replication are not clearly described. Please specify the number of biological and technical replicates for the field experiments and whether the trials were repeated across multiple seasons or locations.

Minor Comments:

Language and Clarity:

The manuscript would benefit from professional English editing to improve clarity and flow. Some sentences are overly long or awkwardly phrased.

Figure and Table Quality:

Some figures (e.g., Fig. 1–4) are difficult to interpret due to overlapping labels and unclear legends. Please improve the resolution and organization of figures.

Table 3 (primers) is in Chinese. All content should be in English for international readability.

Introduction and Discussion:

The introduction could better contextualize the study within the broader literature on ABA and spikelet dynamics.

The discussion should more directly link the molecular findings (e.g., up-regulation of PYL4, down-regulation of PP2C) to the physiological observations.

Reference Formatting:

Some references are incomplete or inconsistently formatted (e.g., missing titles, journal names not abbreviated). Please ensure all references follow PLOS ONE guidelines.

Conclusion:This manuscript presents valuable insights into the role of ABA in rice spikelet closure and has potential applications in hybrid seed production. With revisions addressing the above points, it would be suitable for publication in PLOS ONE.

Reviewer #2: The manuscript "Abscisic acid positively regulates rice spikelet closure " is a well thought research reaching to a conclusion that reflects solution to the concern raised on spikelet closure. However, there are several issues that must be taken care:

1.I struggled to find in the introduction section of the manuscript why the authors chose to use combination of ABA and MeJA used for spraying. Ultimately I found the answer in the methodology section (L197), which in fact should have been stated in the introduction section.

2. if both Jasmonic acid (JA) and Methyl Jasmonate (MeJA) can promote the opening of mature rice spikelets, why only MeJA has been considered for the study?

3. The authors have considered 3 fertile and 3 rice varieties were considered, but the results obtained for them have not been discussed with regard to any difference the closure of stomata among them. Thus the reason of considering 3 rice varieties in both the sterile and fertile group remained unaddressed.

4. All the figures in their right panel need to state in the caption the 'alphabets' used against the top of each column.

5. The discussion section is without citation of any figures. Besides, the discussion section does not seem to interpret the results depicted in all the figures. It appears the discussion is independent of the results.

Reviewer #3: This manuscript addresses the role of abscisic acid (ABA) in regulating rice spikelet closure and explores its potential application in hybrid rice seed production. To address this question, the authors applied exogenous ABA and the ABA biosynthesis inhibitor fluoridone, analyzed transcriptome changes in lodicules, examined the agronomic effects of combined ABA and methyl jasmonate treatment. The transcriptome analysis suggests coordinated regulation of ABA biosynthesis and signaling genes during spikelet movement, and the combined hormone treatment is reported to enhance spikelet closure and hybrid seed yield.

While the topic is agronomically relevant, and the authors have generated a substantial amount of physiological, hormonal, and transcriptomic data, the manuscript in its current form contains fundamental issues in structure, rigor, and presentation. The biological question is not clearly framed, the Results section lacks synthesis and clarity, and key methodological and statistical details are insufficiently described. Addressing these problems would require a comprehensive reworking of the manuscript rather than a standard revision. I therefore recommend rejection, with the suggestion that the authors substantially revise and reorganize the manuscript before considering resubmission to an appropriate journal.

6. PLOS authors have the option to publish the peer review history of their article (what does this mean?). If published, this will include your full peer review and any attached files.

Reviewer #1: No

Reviewer #2: **Yes:** Birendra Prasad Shaw

Reviewer #3: No

---

## [Author Response · Author response to Decision Letter 1]

7 Apr 2026

Dear Editors and Reviewers,

I am Youming Huang,the submitting author of the manuscript "Abscisic acid positively regulates rice spikelet closure" (PONE-D-25-60666).Many thanks to the editor and the reviewers for their dedication to my paper.All authors of this manuscript have made careful and thorough revisions，and have provided responses to every query from the editor and reviewers. I will now present our point-by-point responses:

To editors:

Respond：We have compared our manuscript against PLOS ONE's style requirements one by one, including font, line spacing, file naming, and references. It now meets PLOS ONE's style requirements.

2.In your Methods section, please provide additional information regarding the permits you obtained for the work. Please ensure you have included the full name of the authority that approved the field site access and, if no permits were required, a brief statement explaining why.

Respond：No permits were required, as this study investigated the physiological mechanisms of rice (Oryza sativa ).The study does not pertain to ethical issues, environmental protection, or wildlife collection, and no personal conflict of interest exists.

3. Thank you for stating the following in the Acknowledgments Section of your manuscript: “National Natural Science Foundation Project (31360297)”We note that you have provided funding information that is not currently declared in your Funding Statement. However, funding information should not appear in the Acknowledgments section or other areas of your manuscript. We will only publish funding information present in the Funding Statement section of the online submission form. Please remove any funding-related text from the manuscript and let us know how you would like to update your Funding Statement. Currently, your Funding Statement reads as follows: “Xiaochun Zeng（XCZ）the National Natural Science Foundation Project (31360297) National Natural Science Foundation of China https://www.nsfc.gov.cn/”Please include your amended statements within your cover letter; we will change the online submission form on your behalf.

Respond："'National Natural Science Foundation Project (31360297)' has been deleted from the manuscript.

Please revise the funding statement to“Grant (No:31360297) from the National Natural Science Foundation of China to Xiaochun Zeng”，and kindly add this grant to the online submission form's funding section for me. Thank you!"

Respond：The grant information provided in the 'Funding Information' and 'Financial Disclosure' sections is consistent for this study.

“Grant (No:31360297) from the National Natural Science Foundation of China to Xiaochun Zeng”.Please kindly add this grant to the online submission form's ‘Funding Information’ for me. Thank you!"

5. Thank you for stating the following financial disclosure: “Xiaochun Zeng（XCZ）the National Natural Science Foundation Project (31360297) National Natural Science Foundation of China https://www.nsfc.gov.cn/”

Respond：The funders had no role in study design, data collection and analysis, decision to publish, or preparation of the manuscript.

6. We note that your Data Availability Statement is currently as follows: “All relevant data are within the manuscript and its Supporting Information files.”Please confirm at this time whether or not your submission contains all raw data required to replicate the results of your study. Authors must share the “minimal data set” for their submission. PLOS defines the minimal data set to consist of the data required to replicate all study findings reported in the article, as well as related metadata and methods (https://journals.plos.org/plosone/s/data-availability#loc-minimal-data-set-definition).

Respond：

（1）We hereby certify that the submitted manuscript includes the complete raw data set necessary to reproduce the recults reported in our study.

（2）The “minimal data set ”required to replicate the findings is available in the attached file.

（3）The RNA-seq raw data of this study has not yet been fully analyzed and processed; it remains in the confidential stage and therefore has not been deposited in a public repository. We appreciate your understanding.

7. PLOS requires an ORCID iD for the corresponding author in Editorial Manager on papers submitted after December 6th, 2016. Please ensure that you have an ORCID iD and that it is validated in Editorial Manager. To do this, go to ‘Update my Information’ (in the upper left-hand corner of the main menu), and click on the Fetch/Validate link next to the ORCID field. This will take you to the ORCID site and allow you to create a new iD or authenticate a pre-existing iD in Editorial Manager.

Respond：The corresponding author of this manuscript, Xiaochun Zeng, has an ORCID iD of 0000-0002-0449-0110. Dear Editor, could you please help verify this information in the Editorial Manager system? Thank you!.

Respond：The referee's remarks contain no recommendations regarding the citation of particular prior publications.

To reviewer #1:

Reviewer #1: Overall Recommendation:Major Revision

General Comments:

This study investigates the role of abscisic acid (ABA) in regulating rice spikelet closure using physiological, molecular, and field application approaches. The topic is of significant agronomic importance, especially for hybrid rice seed production. The experimental design is generally sound, and the combination of hormone treatments, RNA-seq, and field trials provides a comprehensive approach. However, several issues need to be addressed before the manuscript can be considered for publication.

Major Comments:

1.Lack of Statistical Details:

While the manuscript mentions statistical tests (e.g., F-tests, multiple comparisons), it does not provide sufficient details on the statistical methods used, such as the specific type of ANOVA, post-hoc tests, or sample sizes for each experiment. Please include these details in the Methods section and ensure that all figures include clear indications of statistical significance (e.g., asterisks with defined p-values).

Respond：

We have made substantial revisions to the details of the statistical methods：

Data analysis was similar to what we used previously. The data in the Figs and tables

were the means and standard deviations of three biological sets of data with each biological set

came from three technical measurements. Mean values within table columns were analyzed

with Duncan new multiple range test. Values in a column with different letters indicate a

significant different difference among the floret stages. Uppercase and lowercase letters repre-

sent significant differences between floret stages at ρ = 0.01 and ρ = 0.05, respectively.

（See manuscript）

2.RNA-Seq Data Availability:

The manuscript states that "all relevant data are within the manuscript and its Supporting Information files." However, RNA-seq data are not typically included in the main text or supplementary files. The raw or processed RNA-seq data should be deposited in a public repository (e.g., NCBI GEO or SRA), and the accession number must be provided.

Respond：We have not deposited the raw or processed RNA-seq data in a public repository.As the RNA-seq raw data of this study have not yet been fully analyzed and processed, they remain in the confidential stage. We appreciate your understanding.

3.Hormonal Interactions:

The study focuses on ABA but also involves MeJA. However, the interaction between ABA and other hormones (e.g., IAA, GA) is only briefly discussed. Given the complexity of hormonal regulation, more discussion or data on how ABA interacts with other hormones during spikelet closure would strengthen the manuscript.

Respond：Given the complexity of hormonal regulation, we have made the following revisions to strengthen the manuscript:

MeJA is currently recognized as the most potent inducer of floret opening, effective in various graminaceous plants including rice, wheat, and sorghum[16]. It promotes floret opening by enhancing ATPase and CAT activities in rice lodicules, facilitating the accumulation of osmotic regulatory substances (such as soluble sugars and ascorbic acid), reducing H2O₂concentration, enabling lodicule water absorption and swelling, and ultimately driving floret opening[9]. The present study demonstrates that exogenous ABA treatment significantly increases the floret closing rate, shortens floret opening duration, and strongly promotes floret closure. Although MeJA and ABA possess independent biosynthetic pathways representing a metabolic parallel relationship, cross-talk exists between them: ABA → SAPK10 (SnRK2 family kinase) → bZIP72 (transcription factor) → AOC gene expression → increased JA/MeJA synthesis. Upon autophosphorylation, SAPK10 phosphorylates bZIP72, enhancing its protein stability and DNA-binding capacity, thereby promoting AOC promoter activity[53]. Future research should further elucidate the spatiotemporal specificity mechanisms underlying ABA-JA metabolic cross-regulation during floret opening, particularly the activity status of the SAPK10-bZIP72-AOC module in lodicule cells.

（See manuscript）

[53]Cui Huan, Gao Qiaoli, Luo Lixin, Yang Jing, Chen Chun, Guo Tao, Liu Yongzhu,Huang Yongxiang, Wang Hui, Chen Zhiqiang, Xiao Wuming.Transcriptome Analysis of Hormone Signal Transduction and Glutathione MetabolicPathway in Rice Seeds at Germination Stage[J].Chinese Journal of Science, 2021, 35(6): 554－564.

4.Mechanistic Insight:

While RNA-seq identified DEGs related to ABA synthesis and signaling, functional validation (e.g., mutant analysis, transgenic plants) is lacking. The authors should acknowledge this limitation and discuss how future studies could address it.

Respond：We have added this discussion in the revised manuscript.

Yes, through RNA-seq sequencing, we identified a large number of differentially expressed genes (DEGs) related to ABA biosynthesis and degradation as well as ABA signal transduction, such as xanthoxin dehydrogenase genes, phosphatase genes in the PP2C family, and so on. In this study, we did not validate their functions (e.g., mutant analysis, transgenic plants), which is a limitation of our work. In future research, we will conduct in-depth studies by combining proteomics.

5.Field Application Data:

The field trial results are promising, but the sample size and replication are not clearly described. Please specify the number of biological and technical replicates for the field experiments and whether the trials were repeated across multiple seasons or locations.

Respond：Thank you for your thorough and careful review.

This has been modified to clarify the sample size and reproducibility（See manuscript）.

The experiment was not conducted across multiple seasons or locations, because the aim of this study was to explore the mechanism of ABA-mediated rice floret closure. In the future, we will increase replication across multiple seasons and locations to prepare for the practical application of ABA and MeJA in hybrid rice seed production.

Minor Comments:

Language and Clarity:

The manuscript would benefit from professional English editing to improve clarity and flow. Some sentences are overly long or awkwardly phrased.

Respond：We have carefully and thoroughly polished the entire manuscript, including word choice, sentence structure, grammar, and sentence length, to better conform to the conventions of scientific writing.

（See manuscript）

Figure and Table Quality:

Some figures (e.g., Fig. 1–4) are difficult to interpret due to overlapping labels and unclear legends. Please improve the resolution and organization of figures.

Table 3 (primers) is in Chinese. All content should be in English for international readability.

Respond：

（1）Regarding the issue of overlapping labels and unclear legends, we have improved the resolution of the figures and enhanced their organization.

（2）The Chinese phrases have been translated into English. (See manuscript:序号--No. 基因ID--Gene ID 正向引物--Forward primer 反向引物--Reverse primer)

Introduction and Discussion:

The introduction could better contextualize the study within the broader literature on ABA and spikelet dynamics.

Respond： The introduction has been revised to better contextualized the study within the literature on ABA and spikelet dynamics.

To verify the effect of ABA in promoting rice spikelet closure, and more importantly, to explore its mechanism and identify additional regulatory techniques for rice spikelet closure, this study tested the effects of ABA and the ABA biosynthesis inhibitor FL on rice spikelet closure using three fertile and three sterile rice lines, monitored the dynamic levels of endogenous ABA in lodicules during the closure process, and further analyzed the differential expression of key enzyme genes involved in ABA biosynthesis and degradation and genes related to ABA signal transduction in lodicules at the transcriptional level during spikelet closure.

（See manuscript）

The discussion should more directly link the molecular findings (e.g., up-regulation of PYL4, down-regulation of PP2C) to the physiological observations.

Respond：We have added the following content in the discussion:

During the critical transition period from floret opening to closing, the β-carotene hydroxylase, the FAD-dependent urate hydroxylase, ZEP, NCED, and the xanthinaldehyde dehydrogenase genes were all up-regulated, indicating elevated endogenous ABA levels in the floret, which is consistent with our measured results (Figure 5). Concurrently,the ABA receptor PYL4 was up-regulated, whereas three PP2C family phosphatases—PP2C66, PP2C8, and PP2C30 were down-regulated,and the SAPK6 from the SnRK2 family was up-regulated(Figure 7, Table 2). This expression pattern is consistent with the dual negative regulatory

---

## [Decision Letter · Decision Letter 1]

28 Apr 2026

Abscisic acid positively regulates rice spikelet closure

PONE-D-25-60666R1

Dear Dr. Huang,

We’re pleased to inform you that your manuscript has been judged scientifically suitable for publication and will be formally accepted for publication once it meets all outstanding technical requirements.

Kind regards,

Keqiang Wu, Ph.D

Academic Editor

PLOS One

Additional Editor Comments (optional):

Reviewers' comments:

Reviewer's Responses to Questions

**Comments to the Author**

1. If the authors have adequately addressed your comments raised in a previous round of review and you feel that this manuscript is now acceptable for publication, you may indicate that here to bypass the “Comments to the Author” section, enter your conflict of interest statement in the “Confidential to Editor” section, and submit your "Accept" recommendation.

Reviewer #1: All comments have been addressed

2. Is the manuscript technically sound, and do the data support the conclusions?

Reviewer #1: Yes

3. Has the statistical analysis been performed appropriately and rigorously?

Reviewer #1: Yes

4. Have the authors made all data underlying the findings in their manuscript fully available?

Reviewer #1: Yes

5. Is the manuscript presented in an intelligible fashion and written in standard English?

Reviewer #1: Yes

6. Review Comments to the Author

Reviewer #1: You have responded to all the questions and made appropriate modifications on the vast majority of non principled issues. The manuscript has significantly improved in terms of language, statistical methods, charts, literature, introduction, and discussion structure and logic. RNA seq data is not publicly available. This is something that must be resolved. The author needs to immediately prepare to upload the data to public databases such as GEO or SRA. If insisting on not disclosing, extremely special and sufficient reasons need to be given in the response (such as involving unpublished patent applications or ethical/legal restrictions), but "incomplete analysis" and "confidentiality stage" are usually not accepted.

7. PLOS authors have the option to publish the peer review history of their article (what does this mean?). If published, this will include your full peer review and any attached files.

Reviewer #1: No

---

## [Editor Report · Acceptance letter]

PONE-D-25-60666R1

PLOS One

Dear Dr. Huang,

I'm pleased to inform you that your manuscript has been deemed suitable for publication in PLOS One. Congratulations! Your manuscript is now being handed over to our production team.

Kind regards,

on behalf of

Professor Keqiang Wu

Academic Editor

PLOS One